# Identifying Inhibitors of −1 Programmed Ribosomal Frameshifting in a Broad Spectrum of Coronaviruses

**DOI:** 10.3390/v14020177

**Published:** 2022-01-18

**Authors:** Sneha Munshi, Krishna Neupane, Sandaru M. Ileperuma, Matthew T. J. Halma, Jamie A. Kelly, Clarissa F. Halpern, Jonathan D. Dinman, Sarah Loerch, Michael T. Woodside

**Affiliations:** 1Department of Physics, University of Alberta, Edmonton, AB T6G 2E1, Canada; munshi1@ualberta.ca (S.M.); kneupane@ualberta.ca (K.N.); ileperum@ualberta.ca (S.M.I.); mhalma@ualberta.ca (M.T.J.H.); 2Department of Cell Biology and Molecular Genetics, University of Maryland, College Park, MD 20742, USA; jkelly22@umd.edu (J.A.K.); chalpern@terpmail.umd.edu (C.F.H.); 3Janelia Research Campus, Howard Hughes Medical Institute, Ashburn, VA 20147, USA; 4Li Ka Shing Institute of Virology, University of Alberta, Edmonton, AB T6G 2E1, Canada

**Keywords:** coronavirus, SARS-CoV-2, programmed ribosomal frameshifting, translation, therapeutics

## Abstract

Recurrent outbreaks of novel zoonotic coronavirus (CoV) diseases in recent years have highlighted the importance of developing therapeutics with broad-spectrum activity against CoVs. Because all CoVs use −1 programmed ribosomal frameshifting (−1 PRF) to control expression of key viral proteins, the frameshift signal in viral mRNA that stimulates −1 PRF provides a promising potential target for such therapeutics. To test the viability of this strategy, we explored whether small-molecule inhibitors of −1 PRF in SARS-CoV-2 also inhibited −1 PRF in a range of bat CoVs—the most likely source of future zoonoses. Six inhibitors identified in new and previous screens against SARS-CoV-2 were evaluated against the frameshift signals from a panel of representative bat CoVs as well as MERS-CoV. Some drugs had strong activity against subsets of these CoV-derived frameshift signals, while having limited to no effect on −1 PRF caused by frameshift signals from other viruses used as negative controls. Notably, the serine protease inhibitor nafamostat suppressed −1 PRF significantly for multiple CoV-derived frameshift signals. These results suggest it is possible to find small-molecule ligands that inhibit −1 PRF specifically in a broad spectrum of CoVs, establishing frameshift signals as a viable target for developing pan-coronaviral therapeutics.

## 1. Introduction

The 21st century has seen a series of public health emergencies caused by zoonotic coronavirus (CoV) diseases: the SARS epidemic in 2002–2003, periodic MERS outbreaks since 2012, and the ongoing COVID-19 pandemic [1]. Given ever-increasing human contact with major CoV reservoirs such as bats [2] due to habitat encroachment and climate change, novel CoV diseases will likely continue to emerge in the near future, generating new public health challenges. It is therefore urgent to identify anti-viral therapeutics that are effective against a broad spectrum of CoVs, especially CoVs derived from bats, which are thought to have been the source of the previous 21st-century CoV zoonoses and are one of the most likely sources for future novel CoVs [2,3]. To date, however, no drugs proven to be effective against a broad spectrum of CoVs have been identified.

One possible target for developing broad-spectrum CoV therapeutics is a process that plays a key role in gene expression in all CoVs: −1 programmed ribosomal frameshifting (−1 PRF). The proteins needed for transcription and replication of the viral RNA in CoVs are encoded in ORF1b, which is out of frame with respect to ORF1a, and they are only expressed when the ribosome shifts into the −1 reading frame at a specific location in the viral genome [4]. This programmed frameshift is directed by a tripartite signal in the mRNA that consists (from 5′ to 3′) of (i) a heptameric ‘slippery sequence’ where the reading-frame shift occurs, (ii) a ~5–7-nucleotide (nt) spacer, and (iii) a structure in the mRNA that stimulates the frameshift [5,6,7]. Altering the level of −1 PRF by mutations or ligands that interact with the stimulatory structure can significantly attenuate replication of SARS-CoV and SARS-CoV-2 [8,9,10,11], as well as of other −1 PRF-dependent viruses [12,13,14], leading to efforts to find ligands with potential therapeutic activity. An additional benefit of −1 PRF as a drug target is that it is orthogonal and complementary to more standard strategies of targeting viral proteins such as the RNA-dependent RNA polymerase (RdRP) or viral proteases, holding out the promise for combination therapies that could be particularly effective when combining suppression of RdRP expression by −1 PRF inhibition with suppression of RdRP activity.

In the case of CoVs, the stimulatory structure is an RNA pseudoknot, a structure formed from two or more interleaved hairpins in which the loop of one hairpin is base-paired to form the stem of another (Figure 1A). Most drugs target proteins, whose well-defined structures and binding pockets allow for high-affinity and high-specificity binding. Because RNA structures such as pseudoknots also offer complex surfaces with binding pockets for potential ligands, they are also well-suited for druggability [15,16,17]. Indeed, CoV pseudoknots feature an unusual 3-stem architecture that is more complex than the 2-stem architecture typical of most stimulatory pseudoknots [18], as illustrated in recent studies of the SARS-CoV-2 pseudoknot (Figure 1B) [10,19,20,21]; this complexity creates a number of putative binding pockets [22] and reduces the likelihood of off-target binding. Moreover, the pseudoknot sequence is generally highly conserved in CoVs [23], suggesting not only that many CoV pseudoknots may share structural features that could allow for ligands to interact with a broad spectrum of CoV pseudoknots, but also that they may be less susceptible to mutations that induce drug resistance by altering the pseudoknot structure. Drugs targeting −1 PRF in CoVs need not necessarily interact directly with the frameshift signal, however: they might also act more indirectly, for example, by affecting the concerted interplay between the frameshift signal, ribosome, and elongation factors that governs −1 PRF [24,25,26,27].

Several studies have identified small-molecule ligands that modulate −1 PRF in human CoVs. Computational docking against the SARS-CoV pseudoknot found a compound, 2-[{4-(2-methyl-thiazol-4ylmethyl)-[1,4]diazepane-1-carbonyl]-amino}-benzoic acid ethyl ester (hereafter denoted MTDB), that inhibited −1 PRF in SARS-CoV [28,29], SARS-CoV-2 [19], and mutant variants of SARS-CoV-2 [30]; MTDB also suppressed replication of SARS-CoV-2 [10]. More recently, empirical screens for modulation of −1 PRF in SARS-CoV-2 have found a number of compounds that either enhance or suppress −1 PRF [11,31,32], including merafloxacin, a fluoroquinolone antibacterial that was also effective at suppressing viral replication and showed resistance to natural mutants of the pseudoknot [11]. Anti-sense oligomers have also been explored for modulating −1 PRF in SARS-CoV [33], MERS-CoV [34], and SARS-CoV-2 [21]. Intriguingly, a novel small-molecule compound was recently found to inhibit −1 PRF in all three of these human CoVs [35]. However, little has been reported about frameshifting or frameshift inhibitors in bat CoVs, which are the likely source of five of the seven existing human CoVs. Although many bat-CoV sequences have been determined and analyzed to identify putative −1 PRF signals [23], frameshifting by these −1 PRF signals has not yet to our knowledge been confirmed experimentally. Furthermore, no pseudoknot structures from bat CoVs have been reported to date, nor have any studies of inhibitors targeting frameshifting in bat CoVs.

In this study we examine −1 PRF in bat CoVs by comparing it to −1 PRF in bat-derived human CoVs such as SARS-CoV-2 and MERS-CoV. The activity of the putative −1 PRF signals from a panel of four CoVs chosen to be representative of the range of sequences found to date across bat CoVs was first confirmed using a dual-luciferase assay. To augment the pool of potential −1 PRF inhibitors, we screened a library of FDA-approved drugs for activity against −1 PRF in SARS-CoV-2. Choosing four of the hits from this screen as well as two compounds previously confirmed to inhibit −1 PRF in SARS-CoV-2 (MTDB and merafloxacin), we then tested their effects on −1 PRF induced by a panel of six frameshift signals from CoVs, obtained from four bat CoVs as well as MERS-CoV and SARS-CoV-2. −1 PRF could be inhibited substantially for each of the bat-CoV frameshift signals, suggesting that they can all be targeted therapeutically. Moreover, several of these compounds had moderate to strong activity against more than one CoV; one, nafamostat, was active to some degree against all of the CoV frameshift signals tested. These results suggest that CoV frameshift signals are a viable target for developing novel broad-spectrum anti-CoV therapeutics.

## 2. Results

### 2.1. Identifying Five Distinct Clusters of −1 PRFsignals in Bat CoVs

A multiple sequence alignment of 959 bat-CoV sequences from the NCBI virus database [36] was performed to build a panel of representative bat-CoV pseudoknots. Of these sequences, 48 had significant coverage of the frameshift region. Calculating pairwise genetic distances between the frameshift signals, we used a multidimensional scaling algorithm [37] to collapse the genetic distances into a 2-dimensional display and thereby identified five distinct clusters. These clusters are illustrated in Figure 1C: one (cluster 5) is closely related to SARS-CoV and SARS-CoV-2, and another (cluster 2) is closely related to MERS-CoV, whereas the other three clusters are either more distantly related or unrelated to human CoVs. In order to sample the breadth of sequence diversity and hence concomitant structural diversity among the frameshift signals, the testing panel was chosen to include one representative sequence from each of clusters 1 through 4 (Figure 1C, cyan), in addition to SARS-CoV-2 from cluster 5 (SARS-like cluster) and MERS-CoV from cluster 2 (Figure 1C, red). The choices for bat CoVs, denoted by their NCBI virus accession numbers, were: KU182958, a beta-CoV isolated from the fruit bat *Rousettus leschenaultii* (cluster 1); LC469308, a beta-CoV isolated from the vesper bat *Vespertilio sinensis* (cluster 2); KY770854, an alpha-CoV isolated from the horseshoe bat *Rhinolophus macrotis* (cluster 3); and KF294282, an alpha-CoV isolated from the bent-wing bat *Miniopterus schreibersii* (cluster 4). The sequences used for each of the frameshift signals are listed in Appendix A.

To confirm that the putative bat-CoV frameshift signals did actually induce −1 PRF and to assess the efficiency with which they did so, we measured the basal efficiency of −1 PRF stimulated by each of the six frameshift signals from the testing panel using cell-free translation of a dual-luciferase reporter system [30,38]. The reporter system consisted of the *Renilla* luciferase gene in the 0 frame upstream of the firefly luciferase gene in the −1 frame, with the two separated by the CoV frameshift signal in the 0 frame. The −1 PRF efficiency was obtained from the ratio of luminescence emitted by the two enzymes, as compared to controls with 100% and 0% firefly luciferase read-through. Every putative frameshift signal did indeed stimulate −1 PRF, but the efficiency varied considerably: the two bat beta-CoV frameshift signals stimulated −1 PRF with ~40% efficiency, comparable to SARS-CoV-2 and MERS-CoV, but the two bat alpha-CoVs did so at noticeably lower levels of ~9% and 25% (Figure 2).

### 2.2. Identifying Small-Molecule Modulators of SARS-CoV-2 −1 PRF

We next investigated the effects of putative small-molecule inhibitors on −1 PRF efficiency for each frameshift signal in the panel. To identify additional inhibitors beyond those reported previously in the literature, we screened a library of 1814 FDA-approved drugs to test their ability to modulate SARS-CoV-2 −1 PRF, using a cell-free dual-luciferase assay optimized for high-throughput screening (Appendix A). Z′ factors of 0.67–0.82 for each plate suggested high quality and reproducibility of the assay [39]. After eliminating compounds that abolished translation and selecting only those that modulated −1 PRF by at least 30% (Appendix A), with a signal at least two standard deviations above background, the initial screen identified 24 hits that had little to no effect on general protein translation (less than two-fold), as indicated by changes in the *Renilla* luminescence levels, and another 18 that also altered general translation levels (Appendix A). Hits from the initial screen were further validated by assaying them again in triplicate, confirming at least a 25% change in −1 PRF for eight of the compounds (Figure 3A). To rule out potential false positives arising from selective inhibition of firefly luciferase, luciferase activities were measured by adding the compounds after translation had been completed but before luminescence was quantified. This process revealed that six of the drugs inhibited −1 PRF, whereas two enhanced it. Nafamostat, a protease inhibitor used as an anticoagulant that also has anti-viral properties and is under clinical investigation for use against COVID-19 [40,41], stood out as the strongest inhibitor.

### 2.3. Identifying Pan-CoV −1 PRF Inhibitors

From these screening results, we selected four of the inhibitors to test their effectiveness against different CoV −1 PRF signals in the cell-free assay: nafamostat, the strongest of the inhibitors; abemaciclib and palbociclib, which are CDK4/6 kinase inhibitors approved for use against breast cancer [42,43]; and valnemulin, as a representative of an antibiotic approved for veterinary use [44]. In each case, IC_50_ values for inhibition of −1 PRF (Table 1) were found from measuring dose–response curves with the cell-free assay (Appendix A). In addition to the cell-free assay, −1 PRF suppression was confirmed for each compound by a cell-based assay using A549 human lung epithelial cells transfected with a bi-fluorescent frameshifting reporter system (Figure 3B). Two compounds previously shown to inhibit −1 PRF by the SARS-CoV-2 frameshift signal were also included in the set of compounds to be tested: MTDB, which was first identified as an inhibitor for SARS-CoV [28,29] before being shown to be active against SARS-CoV-2 [19,30], and merafloxacin, an experimental fluoroquinolone antibiotic shown to be active against SARS-CoV-2 [11]. The structures of each of the compounds used in the tests are shown as insets in Figure 4. For each compound, the percent change in −1 PRF induced by a final ligand concentration of 20 μM was measured for each frameshift signal in the panel. The specificity for CoV frameshift signals was also assessed by measuring the effects of each compound on −1 PRF stimulated by two control frameshift signals: that from pea enation mosaic virus-1 (PEMV1) [45], to test if the compounds interact with a standard but unrelated 2-stem H-type pseudoknot; and that from HIV-1, which involves a simple hairpin [46,47] and hence tests non-specific interactions with duplexes.

First considering the results for MTDB (Figure 4A), we found that it was most effective at inhibiting −1 PRF in SARS-CoV-2 (~55% decrease); it still induced some modest inhibition for the two alpha-CoVs, KY770854 and KF294282 from clusters 3 and 4 (~20% decrease), as well as for KU182958 from cluster 1 (~14% decrease), but it did not inhibit −1 PRF at all in the beta-CoVs from cluster 2, LC469308 and MERS-CoV. The effects of valnemulin (Figure 4B) followed a somewhat similar pattern: most inhibitory for SARS-CoV-2 (~35% decrease in −1 PRF), modest to minimal inhibition for the two alpha-CoVs (~10–25% decrease), and no discernable effect on the remaining beta-CoVs. Abemaciclib (Figure 4C) and palbociclib (Figure 4D) showed a different pattern of inhibition: they were both most effective against KU182958 from cluster 1 and KF294282 from cluster 4 (~45–60% decrease in −1 PRF), had a more modest effect against SARS-CoV-2 (~30%), and were minimally effective or ineffective against the remainder. Merafloxacin (Figure 4E) was most effective against SARS-CoV-2, MERS-CoV, and KY770854 from cluster 3 (~40–55%), modestly effective against KF294282 from cluster 4 and LC469308 from cluster 2 (~20–25%), but ineffective against KU182958 from cluster 1. Finally, nafamostat (Figure 4F) stood out as having an effect for all the frameshift signals tested: it was most effective at inhibiting −1 PRF in SARS-CoV-2, KU182958 from cluster 1, KY770854 from cluster 3, and KF294282 from cluster 4, leading to similar decreases (~40–50%) for all four, and least effective against MERS-CoV and LC469308 from cluster 2 (~20% decrease). For every compound tested, the effects on −1 PRF were consistent with zero for the frameshift signals from HIV and PEMV1 that were used as controls for specificity, with two exceptions that showed a small effect: MTDB with HIV (8 ± 3% decrease) and merafloxacin with PEMV1 (16 ± 5%).

## 3. Discussion

The results represent, to our knowledge, the first measurements of −1 PRF induced by bat coronavirus frameshift signals. They confirm that there is indeed a programmed frameshift at the expected site between ORF 1a and 1b in each of the viruses chosen to represent the five different clusters of bat CoVs. Given that the representative frameshift signals from all clusters except cluster 3 stimulated −1 PRF with efficiency of 25–50%, these findings suggest that most bat CoVs feature relatively high levels of frameshifting, values that are within the range that has been reported previously for several other coronaviruses [19,34,48,49]. The frameshift signal from KY770854 (cluster 3) was a notable outlier; however, stimulating −1 PRF with only 9% efficiency, distinctly below the range typical of CoVs. The origin of this difference is unclear, but it might arise from differences in the pseudoknot structure: the pseudoknot from KY770854 is predicted to contain four stems, rather than the three stems found more commonly in CoV pseudoknots and seen or predicted in all the other members of the panel (Appendix A). Another possibility is that −1 PRF efficiency for this virus might depend on other regulatory elements not included in the assay.

Examining the effects of the inhibitors, the results of this study show that −1 PRF could be suppressed moderately to strongly (~25–60% decrease) for each of the representative frameshift signals in the testing panel by at least one of the inhibitors: (i) KU182958 from cluster 1 by abemaciclib, palbociclib, and nafamostat; (ii) LC469308 and MERS-CoV from cluster 2 by merafloxacin; (iii) KY770854 from cluster 3 by nafamostat and merafloxacin; (iv) KF294282 from cluster 4 by palbociclib, nafamostat, abemaciclib, and valnemulin; and (v) SARS-CoV-2 from cluster 5 (SARS-like cluster) by all of the inhibitors tested. These results provide proof-of-principle that effective small-molecule inhibitors can be found for each of these frameshift signals, and they suggest that the same is likely true for the full range of bat-CoV frameshift signals that these clusters represent.

Several of the inhibitors studied here were moderately to strongly effective against the representatives from more than one CoV cluster: abemaciclib and palbociclib against clusters 1, 4, and 5; valnemulin against clusters 4 and 5; merafloxacin against clusters 2, 3, and 5; and nafamostat against clusters 1, 3, 4, and 5. It was previously noted that merafloxacin was effective at inhibiting −1 PRF in human beta-CoVs but not human alpha-CoVs [11]. However, our results show that this pattern does not extend to bat CoVs, as merafloxacin was found to be quite effective (~45% inhibition) against at least one of the alpha-CoVs, KY770854 from cluster 3. Indeed, there were no obvious patterns in terms of effectiveness against alpha-CoVs versus beta-CoVs for any of the inhibitors active against multiple frameshift signals: in every case, they were effective against at least one alpha-CoV and one beta-CoV, suggesting that the family to which the CoV belongs has little influence on the effectiveness of any of these inhibitors.

The results for nafamostat are particularly interesting, for several reasons. First, nafamostat had a broad effect, inhibiting −1 PRF to some degree for all the CoV frameshift signals tested. Second, this breadth was coupled with effectiveness: it induced ~40% inhibition or more for 4 of the 5 clusters of frameshift signals. Third, it had no effect on the non-CoV −1 PRF signals used as negative controls, arguing that it acts specifically in a way that involves the frameshift signal, rather than through other factors that influence the ability of the ribosome to maintain reading frame independent of the −1 PRF signal. These results suggest nafamostat as a lead compound for development of a drug against a very broad spectrum of CoVs, supporting the notion that targeting −1 PRF holds promise for developing broad-spectrum CoV therapeutics. Intriguingly, nafamostat is in clinical trials for treating COVID-19 as a protease inhibitor; it was previously shown to inhibit S-protein mediated membrane fusion and viral infectivity for MERS-CoV [50] and SARS-CoV-2 [40,41]. However, its use as a protease inhibitor at concentrations 1000-fold lower than used here [40] suggests that any contribution to its effectiveness from −1 PRF inhibition is likely minimal.

Finally, we note that while the in vitro setup of our experimental design avoids off-target effects, for example, on ribosome biogenesis, the mechanisms of action of the −1 PRF inhibitors identified here remain unclear. MTDB was found to bind the stimulatory pseudoknot from SARS-CoV, reducing its conformational heterogeneity [29], a feature that is linked to the efficiency with which −1 PRF is stimulated [46,51,52,53,54]. A more recent study found a predictive correlation between MTDB binding and conformational heterogeneity of the SARS-CoV pseudoknot under tension [55], within the range of forces applied to the mRNA by the ribosome during translation and frameshifting [56,57], suggesting that −1 PRF inhibitors may act in part by reducing the conformational heterogeneity of the stimulatory structure. Given the ability of CoV pseudoknots to form multiple folds, some even with different topologies [4,10,20,21,58], it is plausible that inhibitors could stabilize specific folds preferentially, reducing the −1 PRF efficiency. Notably, all inhibitors are small molecules with features that are typical for RNA-binding molecules, including aromatic moieties for π-stacking and polar moieties for interactions with RNA bases and phosphate backbones. All newly identified molecules are of similar size as MTDB, suggesting the possibility that they too bind to the pseudoknot at a similar site. It is also possible that inhibitors may act by modulating interactions between the pseudoknot and the ribosome, given that cryo-EM models of the SARS-CoV-2 pseudoknot complexed with a stalled ribosome identified specific contacts between the two that were found to be important for high −1 PRF efficiency [10]. The mechanism of inhibition might also involve other effects on the machinery of translation, such as interactions with the ribosome or elongation factors that alter the dynamics involved in changing reading frame [24,25,26,27], but the specificity of the inhibitors for certain sets of −1 PRF signals that differ only in their stimulatory structure indicates that any such mechanism would still require some form of interaction with the CoV pseudoknot. Future work may clarify the detailed mechanism of action of −1 PRF inhibitors such as these.

## 4. Methods

Clustering of bat coronavirus genomes: All known bat coronavirus sequences were downloaded on 30 April 2020 as an alignment from the NCBI Virus database. These were imported into Geneious Prime version 2020.1.2 (Biomatters, San Diego, CA, USA), where the frameshift region was extracted. Only sequences with coverage in the frameshift region were retained for clustering. A phylogenetic tree was created based on intersequence distances at the frameshift site, revealing five clusters (Figure 1C). Secondary structural predictions were performed on the pKiss [59] and Hotknots [60,61] web servers using default settings, with the proposed structures (Appendix A) selected as the lowest-energy consensus results homologous to the SARS-CoV and SARS-CoV-2 pseudoknots.

Preparation of mRNA constructs: (1) Constructs for testing inhibitors. To measure −1 PRF efficiency in the panel of CoV frameshift signals for testing inhibitors, we used a dual-luciferase reporting system based on a plasmid containing the sequence for *Renilla* luciferase and the multiple cloning site (MCS) from the plasmid pMLuc-1 (Novagen, Madison, WI, USA) upstream of the firefly luciferase sequence in the plasmid pISO (addgene), as described previously [19,30]. The frameshift signals for all CoVs in the testing panel as well as the two viruses used as negative controls (HIV-1 and PEMV1) were cloned into the MCS between the restriction sites PstI and SpeI. Three different types of constructs were made for each CoV frameshift signal. First, a construct for assaying −1 PRF efficiency was made: it contained the frameshift signal with slippery sequence (UUUAAAC) and pseudoknot, and the downstream firefly luciferase gene was placed in the −1 frame so that its expression was dependent on −1 PRF. Note that while an upstream attenuator hairpin has been identified in the SARS-CoV-2 frameshift signal [19], no comparable attenuators have been confirmed in the other CoV frameshift signals used here; to ensure consistency across the testing panel, we therefore omitted the attenuator from the SARS-CoV-2 frameshift reporter construct. Next, two controls from this construct were made: (i) a negative control to measure the background firefly luciferase luminescence (0% firefly luciferase read-through), where the slippery sequence was mutated to include a stop codon (UUGAAAC); and (ii) a positive control to measure 100% firefly luciferase read-through, where the slippery sequence was disrupted (UAGAAAC) and the firefly luciferase gene was shifted into the 0 frame. Sequences of frameshift signals for all constructs are listed in Appendix A. Transcription templates were amplified from these plasmids by PCR, using a forward primer that included the T7 polymerase sequence as a 5′ extension to the primer sequence [19,30]. The mRNAs for dual-luciferase measurements were then produced from the transcription templates in-vitro using the MEGAscript T7 transcription kit (Invitrogen, Waltham, MA, USA) and purified using the MEGAclear transcription clean-up kit (Invitrogen). All mRNAs were polyadenylated (including 30 A’s at the 3′ end) but uncapped before translation.

(2) Constructs for library screening. The dual-luciferase reporter system used for screening the drug library was based on previously described plasmids [19]. Briefly, the frameshifting element from SARS-CoV-2—encompassing the attenuator hairpin, slippery sequence, spacer, and pseudoknot—was placed between the sequences for *Renilla* and firefly luciferases. For the construct used to measure −1 PRF, *Renilla* luciferase was in the 0 frame and firefly luciferase in the −1 frame. To measure 100% read-through, the firefly luciferase gene was placed in the 0 frame and the slippery sequence was disrupted by mutation as above. For the negative control, equivalent to 0% −1 PRF, silent mutations were introduced into the slippery sequence (CCUCAAC) that left the encoded protein sequence unchanged. Transcription templates were amplified from the plasmids by PCR and transcribed in vitro with the HiScribe T7 High Yield RNA synthesis kit (New England Biolabs, Ipswich, MA, USA). To obtain sufficient amounts of mRNA, reactions were scaled up five-fold compared to the manufacturer’s protocol. To remove rNTPs and polymerase, mRNA was purified using weak anion exchange as described [62]. The mRNA was not polyadenylated and was uncapped before translation.

(3) Construction of bi-fluorescent reporter for cell-based assays: The reporter plasmid pJD2261 was constructed by cloning a gBlock (Integrated DNA Technologies, Coralville, IA, USA) encoding AcGFP, the HIV-1 −1 PRF signal, mCherry, and insulator A2 described previously [63] into PstI-digested pUC19 (Genbank accession L09137). Next, the CMV promoter from pJD2044 was inserted into KpnI/PstI-digested fluorescent reporter. The 0-frame control reporter was made by digesting pJD2261 with SalI and BamHI to remove the HIV-1 frameshift insert, gel-purifying the result, then ligating a DNA oligonucleotide insert containing an α-helix spacer [64], with modified ends containing SalI and BamHI restriction sites. The SARS-CoV-2 frameshift reporter was made by digesting pJD2514 [19] using SalI and BamHI, gel-purifying the SARS-CoV-2 −1 PRF insert, and then ligating it into SalI/BamHI-digested pJD2261 vector. All reporter plasmids were sequence-verified. The oligonucleotide sequences for cloning are listed in Appendix A.

Dual-luciferase assays of −1 PRF in panel of CoV frameshift signals: The −1 PRF efficiency was measured for each member of the panel using a cell-free dual-luciferase assay [38]. Briefly, for each construct, 1.2 µg of mRNA transcript was heated to 65 °C for 3 min and then incubated on ice for 2 min. The mRNA was added to a solution mixture containing amino acids (10 µM Leu and Met, 20 µM all other amino acids), 17.5 µL of nuclease-treated rabbit reticulocyte lysate (RRL) (Promega, Madison, WI, USA), 5 U RNase inhibitor (Invitrogen), and brought up to a reaction volume of 25 µL with water. The reaction mixture was incubated for 90 min at 30 °C. Luciferase luminescence was then measured using a microplate reader (Turner Biosystems, Sunnyvale, CA, USA). First, 20 µL of the reaction mixture was mixed with 100 µL of Dual-Glo Luciferase reagent (Promega) and incubated for 10 min before reading firefly luminescence, then 100 µL of Dual-Glo Stop and Glo reagent (Promega) was added to the mixture to quench firefly luminescence before reading *Renilla* luminescence. The −1 PRF efficiency was calculated from the ratio of firefly to *Renilla* luminescence, *F*:*R*, after first subtracting the background *F*:*R* measured from the negative control (which defines the signal expected for 0% −1 PRF) and then normalizing by *F*:*R* from the positive control (which defines the signal expected for 100% −1 PRF).

To quantify the effects of the inhibitors on −1 PRF efficiency, compounds were added to the reaction volume at a final concentration of 20 μM for each construct (frameshift signals from CoVs, HIV, and PEMV1, as well as all positive and negative controls). Compounds were dissolved in DMSO, leading to a final DMSO concentration in the assays of 1% by volume. The results were averaged from three to six replicates, as described previously [19,30]. The *Renilla* and firefly luciferase luminescence levels were similar for the different frameshift signals. The compounds did not affect the luminescence levels for *Renilla* luciferase, indicating that they did not interfere with translation, but rather their effects were specific to −1 PRF. Note that the inhibition values recorded for SARS-CoV-2 in these assays differed from the results found in Figure 3A owing to different measurement conditions (e.g., full-concentration RRL in Figure 4 vs. 20% RRL in Figure 3A).

For the drug screen, a dual-luciferase assay in RRL was optimized for a 384-well screening format by varying RRL content and salt concentrations. First RRL content was varied while keeping salt concentrations constant at 45 mM KCl, 90 mM KOAc, and 2.6 mM Mg(OAc)_2_ to find the minimal RRL content yielding reasonable signal; next, KCl concentration was varied while keeping KOAc and Mg(OAc)_2_ as before and using 20% RRL to find the KCl concentration yielding maximal luciferase signal; next, Mg(OAc)_2_ concentration was varied while keeping KOAc as before but using optimized RRL and KCl conditions; finally, KOAc concentration was varied while using the optimized concentrations for the other conditions (Appendix AA–D). The optimized conditions (reaction volume of 5 μL containing 20% RRL, with final concentrations of 25 mM potassium chloride, 0.5 mM magnesium acetate, and 65 mM potassium acetate, as well as 1.6 ng/ul mRNA, supplemented with 1 U/μL RNase inhibitor (RNaseIn, Promega), 20 μM amino acid mix (Promega), and 2 mM DL-dithiothreitol) yielded −1 PRF efficiencies comparable to the standard protocol supplied by the manufacturer (Appendix A).

For screening, a commercial library of 1814 FDA-approved drugs (MedChem Express, Cat. No. HY-L022M; details provided in Appendix A) was reformatted into 384-well plates as 50 μM stock solutions in nuclease free-water with or without 1% DMSO, depending on the drug solubility. 1.0 μL of each compound was added to 4 μL of master mix containing all other reagents in 384-well flat white plates (Corning, Corning, NY, USA) on ice, for a final drug concentration of 10 μM. Positive and negative controls, as well as the frameshifting construct for normalization, were included in alternating wells of columns 1, 2, 23, and 24 on each plate, all treated with 1.0 μL of 1% DMSO. Plates were sealed, briefly mixed and centrifuged, and then incubated at 30 °C for 75 min. Reactions were quenched after incubation with 20 μM puromycin, and 20 μL of each luciferase substrate from the Luc-Pair Duo-Luciferase HT Assay kit (GeneCopoeia, Rockville, MD, USA) was added sequentially following manufacturer instructions. Luminescence was measured with 2 s integration times in a microplate reader (Tecan Spark), 20 min after adding each substrate. The addition of reagents to the microplates was timed at each step to match the reading time delays and reading sequence of the microplate reader.

Z′-factors were determined for each plate using positive, negative, and −1 PRF controls as previously described [39]. The normalized −1 PRF inhibition was calculated by setting the average signal of the negative control (slippery-site mutant) to 100% inhibition, the average signal of the positive control (0 frame read-through) to 0%, and normalizing the result for each drug to the drug-free −1 PRF control. All data analysis was carried out in Graphpad Prizm 8 and 9. We tested hits for selective inhibition of firefly luciferase, which will lead to false positives, by incubating reactions without any drug present to produce the ratio of firefly and *Renilla* luciferases expected in the absence of drug, quenching the reaction with puromycin, then adding the compounds and continuing as in the standard protocol (Appendix A). We also ensured hits altered general levels of translation less then two-fold, by comparing the *Renilla* luciferase signals measured with and without compound (Appendix A).

Bi-fluorescence frameshift measurements: A549 (CRL-185) cells were purchased from the American Type Culture Collection (Manassas, VA, USA). Cells were maintained in F-12 K media (Corning) supplemented with 10% fetal bovine serum (Gibco) and 1% penicillin–streptomycin, and grown at 37 °C in 5% CO_2_. Cells were seeded at 1 × 10^5^ cells per well in 12-well plates. Cells were transfected 24 hr after seeding with 0.75 µg of the bi-fluorescent reporter plasmid using Lipofectamine 3000 (Invitrogen) as per manufacturer’s protocol. Cells were treated 24 h after transfection with a final concentration of 10 µM of the designated inhibitor, and then incubated for an additional 24 h. Cells were collected by scraping into phosphate-buffered saline, pelleted by centrifugation, then lysed in Triton lysis buffer (1% Triton, 150 mM NaCl, 50 mM Tris pH 8, 1× Halt protease-inhibitor cocktail (Thermo Fisher Scientific, Waltham, MA USA)). Cell lysates were clarified by centrifugation and then assayed in a clear-bottom black-walled 96-well plate (Grenier Bio-One), quantifying fluorescence using a GloMax microplate luminometer (Promega). Fluorescence levels were measured using the “green” optical kit for mCherry (excitation at 525 nm, emission at 580–640 nm) and the “blue” optical kit for AcGFP (excitation at 490 nm, emission at 510–570 nm). The −1 PRF efficiencies were calculated (after correcting for AcGFP fluorescence bleed-over into the mCherry channel) from the ratio of mCherry to AcGFP fluorescence, *mCh*:*AcG*, by first subtracting the background *mCh*:*AcG* measured from mock transfected cells (empty vector control defining 0% −1 PRF) and then normalizing by the 100% read-through *mCh*:*AcG* from the positive control. Note that −1 PRF values measured in cells often differ between cell lines and from values measured in RRL [65,66,67], accounting for the differences seen with respect to the RRL-based results.

## Figures and Tables

**Figure 1 viruses-14-00177-f001:**
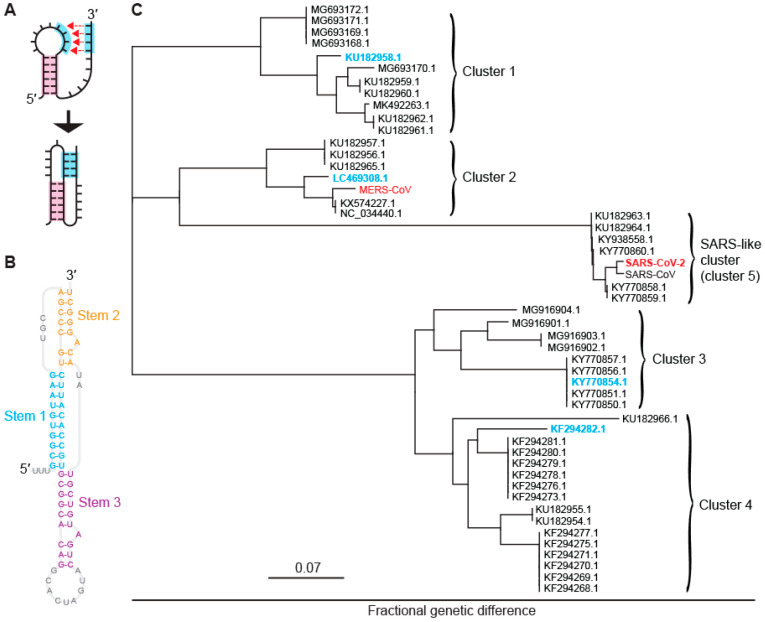
Pseudoknots from bat coronaviruses. (**A**) Pseudoknots form when the unpaired bases in an RNA stem loop pair with another single-stranded segment. (**B**) CoV pseudoknots have a 3-stem architecture, illustrated here for the pseudoknot from SARS-CoV-2. (**C**) Phylogenetic tree showing five clusters of bat and human CoVs with similar pseudoknot sequences. Representatives from each cluster studied are shown here in cyan for bat CoVs and red for human CoVs.

**Figure 2 viruses-14-00177-f002:**
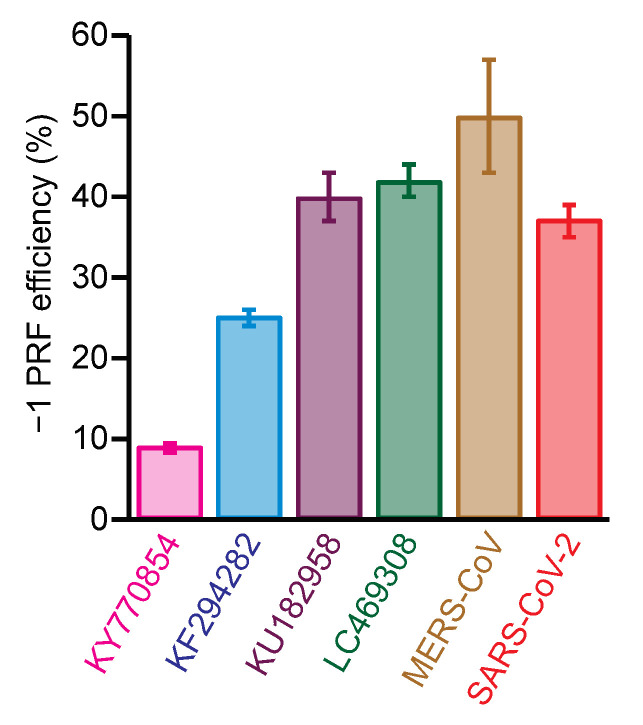
Efficiency of −1 PRF stimulated by bat CoV frameshift signals. The −1 PRF efficiency measured from cell-free dual-luciferase assays is in the range 25–50% typical of CoVs for all except KY770854 (cluster 3). Error bars represent standard error of the mean from 4–17 replicates.

**Figure 3 viruses-14-00177-f003:**
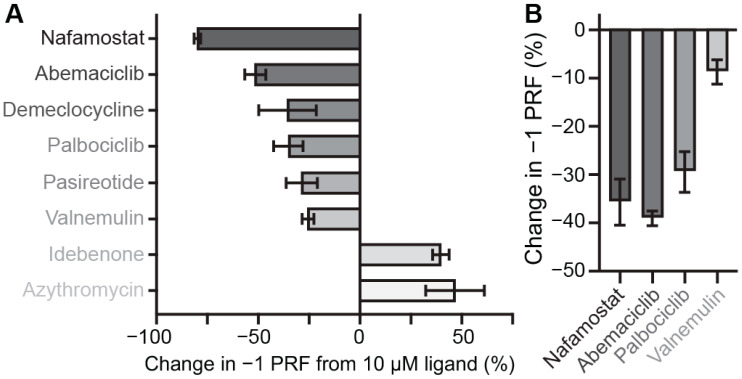
Drugs modulating −1 PRF in SARS-CoV-2 from screening assays. (**A**) Results from screening a library of 1814 FDA-approved drugs using a dual-luciferase reporter measured in vitro in rabbit-reticulocyte lysate; most of the hits inhibited −1 PRF (dark grey), but some enhanced it (light grey). Error bars represent the standard error of the mean from 3 replicates. (**B**) Inhibition of −1 PRF by selected compounds in A549 human lung epithelial cells transfected with a bi-fluorescent reporter system. Error bars represent standard error of the mean from 4 replicates.

**Figure 4 viruses-14-00177-f004:**
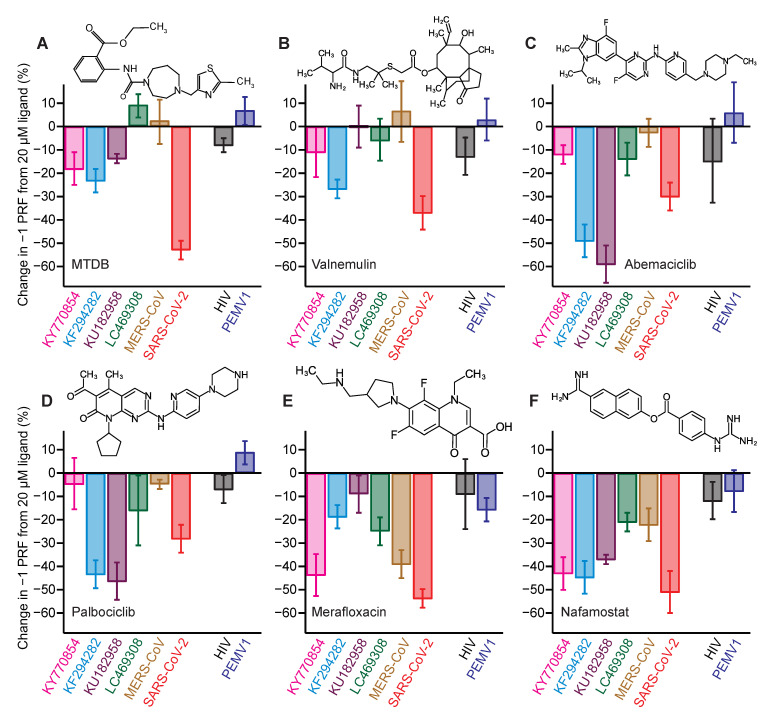
Activity of −1 PRF inhibitors against frameshift signals from different CoVs. (**A**) Change in −1 PRF efficiency compared to basal levels (Figure 2) induced by 20 μM MTDB. Remaining panels show the same for (**B**) valnemulin, (**C**) abemaciclib, (**D**) palbociclib, (**E**) merafloxacin, and (**F**) nafamostat. In each case, results for CoVs are shown on left, results for specificity controls on right. Experiments performed in vitro using dual-luciferase reporter in rabbit reticulocyte lysate. Error bars represent standard error of the mean from 3–6 replicates. Insets: chemical structures of inhibitors.

**Table 1 viruses-14-00177-t001:** −1 PRF inhibition IC_50_ values. IC_50_ values found from fitting dose–response of −1 PRF inhibition for SARS-CoV-2 frameshift signal (Appendix A).

Inhibitor	IC_50_ (μM)
nafamostat	0.5 ± 0.4
abemaciclib	0.6 ± 0.2
palbociclib	0.6 ± 0.3
valnemulin	0.04 ± 0.03

## Data Availability

The data supporting the reported results are provided in the Appendix A.

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
