# Peer review of "Identifying Inhibitors of −1 Programmed Ribosomal Frameshifting in a Broad Spectrum of Coronaviruses"

_viruses, 2022, doi:10.3390/v14020177_

Round 1

Reviewer 1 Report

The manuscript titled “Identifying inhibitors of −1 programmed ribosomal frameshifting in a broad spectrum of coronaviruses “ provides an interesting analysis of the -1 PRF SARS-COV2 as potential drug target for the identification of pan-coronaviruses inhibitors. The workflow, which led to the identification of the six inhibitors, is well described. However, a few critical points need to be addressed before publication.

1)There is not enough information on the similarity and difference of the different coronovavirus frameshifting elements used to assess the activity of the inhibitors. Although the author claimed that the pseudoknot sequence is generally highly conserved in CoVs, even a difference of a few nucleotides can drastically change the RNA structure and the potential binding pockets. The author might consider expanding this section, adding more information on the Frameshifting element structures.

2) The identified drugs showed a high variability activity among the different PRF coronaviruses, suggesting that the drugs have a different affinity against the PRF coronaviruses. Still, the authors did not explain these differences.

3)The sentence: “Notably, all inhibitors are small molecules with features that are typical for RNA-binding molecules, including aromatic moieties for π-stacking and polar moieties for interaction with RNA bases and phosphate backbones” is far too generic, also considering that are FDA approved drugs with well know identified targets. Several studies have reported that the RNA-binding molecules possess specific requirements compared to other inhibitors, such as higher H bond donor, aromatic ring, acceptor, fewer aliphatic compounds, etc. The author might consider analysing the chemical property of the identified inhibitors and checking if they can meet those requirements.

4) There is very little about the mechanism of action of these compounds and their potential interaction

5) The author tested the compounds at a single dose of 20uM. I strongly suggested testing the best compounds at least at different doses to obtain an IC50.

6) There is no information regarding the actual antiviral effect of those drug and their cytotoxicity. This information is very relevant and should be added to the manuscript.

In conclusion, the manuscript could be considered for publication, after some major revisions.

Reviewer 2 Report

-1 Programmed ribosomal frameshifting (-1 PRF) is a translation mechanism to regulate the relative levels of two or more proteins from a single mRNA template. This mechanism is widely used in a variety of viruses, including SARS-CoV-2, a coronavirus that causes the COVID-19 pandemic. Maintaining an appropriate efficiency of -1 PRF is important for viral proliferation. It has thus been a therapeutic target by modulating the -1 PRF efficiency. A key factor to stimulate -1 PRF is a specific secondary structure formed by mRNA. In this report, the authors first constructed an mRNA reporter containing the -1 PRF motif of SARS-CoV-2 and used this reporter to screen a library of FDA-approved drugs. Then, a few top-ranked drugs against SARS-CoV-2 were selected to test their effectiveness on other -1 PRF motifs selected from five representative bat coronaviruses, likely the source for previous and future novel coronaviral diseases. A drug among them was identified to show a broad effect.

Overall, the experimental designs and data are well presented, and the results are clearly explained. This is an interesting paper and merits publication. Here I have only some minor points.

Lines 144-146: “The mRNA for dual-luciferase measurements were then produced from the transcription templates by in-vitro transcription (MEGAclear).” This MEGAclear is a kit for RNA purification, instead of a kit for transcription. In addition, have the in vitro transcribed mRNAs (described here and in the following paragraph) been 5’ capped and/or polyadenylated before the use for translation?

Line 202 and Figure S1: “with final concentrations of 25 mM potassium chloride, …” This 25 mM concentration of KCl was supposed to choose to optimize a dual-luciferase assay. However, this concentration was not done (and appeared not to be an optimal concentration) in Figure S1B. How 25 mM was chosen? In addition, for each of Figure S1A-1D, how the other components (salts, RRL, etc.) were controlled?

Lines 295-299: The authors have tested 1,814 FDA-approved drugs and selected those that modulated (either promoted or suppressed, I guess) -1 PRF by >= 30%. It will be helpful if an overall distribution of the drug effects on -1 PRF modulation is shown (e.g., a histogram plot of drug counts vs. % changes in -1 PRF).

Reviewer 3 Report

I recommend this paper to be published urgently in Viruses considering the significance of the study to identify antiviral candidates against SARS-CoV2.  A similar screening was conducted by Chen el al (reference [31] A drug screening toolkit based on the–1 ribosomal frameshifting of SARS-CoV-2. Heliyon 2020) using the same FDA-approved drug library, but the outcomes are quite different including the activity of known compound MTDB. Please add a convincing comment in the text or evidence, showing that the current assay system is more reliable and reproducible in comparison with their work. 

Reviewer 4 Report

In this paper, the authors constructed a high-throughput assay system to analyze -1PRF, a common and essential mechanism of coronaviruses, and analyzed -1PRF of pathological coronaviruses and bat coronaviruses. The authors performed compound screening using this assay system and found that several compounds, including nafamostat, specifically inhibit -1PRF of coronaviruses. -1PRF is a promising drug target for coronavirus infection. It is also interesting to note that nafamostat, which was found in this analysis, has already been reported to inhibit coronavirus infection as a protease inhibitor. The discovery of these inhibitors will also help to elucidate the mechanism of -1PRF. Therefore, I think this manuscript is acceptable for Viruses when following minor points are fixed.

  1. Z'-factor is useful as a criterion for stability of HTS. Very good Z'-factor was observed in this assay, indicating that the HTS is functional. By describing the signal to background ratio, the researcher can understand more about the stability of the assay system.
  2. The addition of the list of drugs in the compound library used in this paper to supplemental information will help researchers performing similar analyses. MedChem Express Cat. No. HY-L022M contains a variable number of drugs, so the list provided by the manufacturer on website may not match the compounds analyzed by the authors.
  3. It is stated that after the initial screening, the authors evaluated the effects of the candidate compounds on general protein translation and on the activity of firefly luciferase to eliminate false positives. The data on these non-specific effects for the selected inhibitors (in Fig.3) will clarify the -1PRF-specific effects.

Reviewer 5 Report

HIV Fluorescent Control Construct and potentially other constructs
The HIV-1 Control appears to have a non-native sequence preceding the slippery sequence. However, Kim et. Al have demonstrated that the 7nt preceding the slippery sequence can dramatically impact frameshifting efficiencies.(1) Additionally, Leger and coworkers showed that the codon preceding the slippery sequence, i.e. the codon which pairs with the E-site following a slip also has a significant effect on frameshifting.(2)  To our eyes then, the 7nt directly upstream of the HIV-1 slippery sequence do not match the native HIV-1 sequence nor, does the codon directly preceding the slippery sequence. Furthermore, from your description we are unclear as to whether the dual luciferase constructs, beyond CoV-2 as it includes the upstream attenuator, are constructed in a similar manner.

Attenuator Sequences
We appreciate that the authors chose to include the SARS CoV-2 attenuator sequence as a part of their construct. However, we are uncertain from the text whether they used this same attenuator for each CoV tested or only used the attenuator with the Cov-2 construct? Could the authors please clarify this point? 

Frameshifting and Treatment Controls
The authors state “To rule out potential false positives arising from selective inhibition of firefly luciferase, luciferase activities were measured by adding the compounds after translation had been completed but before luminescence was quantified.” From our understanding, common controls for  PRF dual luciferase assay’s include “Empty Vector”  and “Always -1” neither of which appear to be used in this screen as far as we can see.

1.    Kim, Y. G., Maas, S., and Rich, A. (2001) Comparative mutational analysis of cis-acting RNA signals for translational frameshifting in HIV-1 and HTLV-2. Nucleic Acids Res 29, 1125-1131
2.    Leger, M., Dulude, D., Steinberg, S. V., and Brakier-Gingras, L. (2007) The three transfer RNAs occupying the A, P and E sites on the ribosome are involved in viral programmed -1 ribosomal frameshift. Nucleic Acids Res 35, 5581-5592

Round 2

Reviewer 5 Report

Comments were addressed.